# Evaluation of the Neutralizing Antibody STE90-C11 against SARS-CoV-2 Delta Infection and Its Recognition of Other Variants of Concerns

**DOI:** 10.3390/v15112153

**Published:** 2023-10-25

**Authors:** Leila Abassi, Federico Bertoglio, Željka Mačak Šafranko, Thomas Schirrmann, Marina Greweling-Pils, Oliver Seifert, Fawad Khan, Maeva Katzmarzyk, Henning Jacobsen, Natascha Gödecke, Philip Alexander Heine, André Frenzel, Helena Nowack, Stefan Dübel, Ivan-Christian Kurolt, Roland E. Kontermann, Alemka Markotić, Maren Schubert, Michael Hust, Luka Čičin-Šain

**Affiliations:** 1Department of Viral Immunology, Helmholtz Centre for Infection Research, Inhoffenstr. 7, 38124 Braunschweig, Germany; leila.abassi@helmholtz-hzi.de (L.A.); fawad.khan@helmholtz-hzi.de (F.K.); maeva.katzmarzyk@helmholtz-hzi.de (M.K.); henning.jacobsen@helmholtz-hzi.de (H.J.); natascha.goedecke@helmholtz-hzi.de (N.G.); 2Institut für Biochemie, Biotechnologie und Bioinformatik, Technische Universität Braunschweig, Spielmannstr. 7, 38106 Braunschweig, Germany; f.bertoglio@tu-braunschweig.de (F.B.); p.heine@tu-braunschweig.de (P.A.H.); s.duebel@tu-braunschweig.de (S.D.); maren.schubert@tu-braunschweig.de (M.S.); m.hust@tu-braunschweig.de (M.H.); 3Research Department, University Hospital for Infectious Diseases “Dr. Fran Mihaljević”, 10000 Zagreb, Croatia; zmacak@bfm.hr (Ž.M.Š.); ikurolt@bfm.hr (I.-C.K.); alemka.markotic@gmail.com (A.M.); 4YUMAB GmbH, Inhoffenstr. 7, 38124 Braunschweig, Germany; th.schirrmann@yumab.com (T.S.); a.frenzel@yumab.com (A.F.); 5Core Facility of Comparative Medicine, Helmholtz Centre for Infection Research, 38124 Braunschweig, Germany; marina.greweling-pils@helmholtz-hzi.de; 6Institute of Cell Biology and Immunology, University of Stuttgart, 70174 Stuttgart, Germany; oliver.seifert@izi.uni-stuttgart.de (O.S.); helenanowack@gmx.de (H.N.); roland.kontermann@izi.uni-stuttgart.de (R.E.K.); 7School of Medicine, Catholic University of Croatia, 10000 Zagreb, Croatia; 8Faculty of Medicine, University of Rijeka, 51000 Rijeka, Croatia; 9Centre for Individualized Infection Medicine, a Joint Venture of HZI and MHH, 31625 Hannover, Germany

**Keywords:** SARS-CoV-2, monoclonal antibody, Delta variant, mice experiments, intravenous administration, intranasal administration, pseudovirus assay, single mutations, bispecific antibodies

## Abstract

As of now, the COVID-19 pandemic has spread to over 770 million confirmed cases and caused approximately 7 million deaths. While several vaccines and monoclonal antibodies (mAb) have been developed and deployed, natural selection against immune recognition of viral antigens by antibodies has fueled the evolution of new emerging variants and limited the immune protection by vaccines and mAb. To optimize the efficiency of mAb, it is imperative to understand how they neutralize the variants of concern (VoCs) and to investigate the mutations responsible for immune escape. In this study, we show the in vitro neutralizing effects of a previously described monoclonal antibody (STE90-C11) against the SARS-CoV-2 Delta variant (B.1.617.2) and its in vivo effects in therapeutic and prophylactic settings. We also show that the Omicron variant avoids recognition by this mAb. To define which mutations are responsible for the escape in the Omicron variant, we used a library of pseudovirus mutants carrying each of the mutations present in the Omicron VoC individually. We show that either 501Y or 417K point mutations were sufficient for the escape of Omicron recognition by STE90-C11. To test how escape mutations act against a combination of antibodies, we tested the same library against bispecific antibodies, recognizing two discrete regions of the spike antigen. While Omicron escaped the control by the bispecific antibodies, the same antibodies controlled all mutants with individual mutations.

## 1. Introduction

The Coronavirus disease 2019 (COVID-19), caused by the severe acute respiratory-distress syndrome coronavirus 2 (SARS-CoV-2), has been the largest global pandemic in decades. People infected with SARS-CoV-2 suffer from respiratory, cardiovascular, and neurological diseases [1,2]. The symptoms have ranged from asymptomatic to severe symptoms leading to death or Long COVID with neuropsychological syndromes [3]. Since then, new variants emerged, harboring mutations in the spike protein that affect the viability and transmission of the virus. D614G was one of the first mutations that made the virus more infectious. This was followed by several mutations that evolved into new circulating variants of concern that have sustained infectious waves worldwide; Alpha (B1.1.7), Beta (B1.351), Delta (B1.617.2), and Omicron (B1.1.529) [4]. Several studies have reported that the early variants of concern (Alpha, Beta, and Delta), exhibited higher pathogenicity, virus growth in lungs, and mortality than the Omicron sublineages in humans and SARS-CoV-2 animal models [5,6,7], where Delta was particularly pathogenic [8].

Throughout the last few years, numerous therapeutic or prophylactic antiviral interventions have been approved, including numerous vaccines, antiviral drugs, and monoclonal antibodies. While drugs such as Paxlovid have been effective against all evolving variants, vaccines and monoclonal antibodies have lost potency as novel variants with mutated spike (S) proteins have emerged. The most radical escape from antiviral activity was observed among the monoclonal antibodies targeting SARS-CoV-2. Four monoclonal antibodies (bamlanivimab plus etesevimab, casirivimab plus imdevimab, sotrovimab, and bebtelovimab) were authorized as emergency use for therapeutic and prophylactic approaches and worked efficiently against ancestral variants, but were removed from the market due to their failure in neutralizing the Omicron variant or Omicron sublineages. Another monoclonal antibody mix EVUSHELD (tixagevimab + ciglavimab) was granted Emergency Use Authorization (EUA) by the US FDA as a prophylactic treatment and remained efficient against BA.1 to BA.5 variants of Omicron, but had to be retracted from the market because novel SARS-CoV-2 Omicron variants escaped its recognition [9]. Hence, the natural selection of variants avoiding immune recognition has also been a major driver of resistance against monoclonal antibodies used in the therapy of COVID-19. 

STE90-C11 (COR-101) is a previously described monoclonal antibody (mAb) targeting the same RBD region as the ACE2. It protects against the SARS-CoV-2 ancestral strain in vitro and in vivo [10]. In this study, we check this mAb against the Delta variant in vitro and in vivo, and analyze the molecular basis of its failure to recognize the Omicron variant. To this end, we have generated a library of pseudoviruses carrying single amino acid mutations present in the Omicron variant and show which ones are responsible for the failure of recognition against the Omicron variant. This approach is used to analyze the neutralization capacity of a class of bispecific antibodies based on the monoclonal anti-NTD antibody FC05 [11] and anti-RBD P17 [12] against SARS-CoV-2 variants, where we demonstrate how multiple simultaneous mutations in Omicron allowed it to avoid recognition by bivalent antibody formulations. 

## 2. Materials and Methods

### 2.1. Cell Culture and Viruses

All cell lines were maintained at 37 °C and 5% CO_2_. Vero E6 (CRL-1586) and Vero TMPRSS [13] were cultured in Dulbecco’s modified eagle medium (DMEM) supplemented with 10% fetal calf serum (FCS), 2 mM L-glutamine, 100 IU/mL penicillin, and 100 µg/mL streptomycin. A549AT cells expressing ACE2 and TMPRSS2 [14] and CaCo-2 cells (provided by Dr. Denisa Bojkova/Frankfurt) were maintained in Minimum Essential Medium (MEM) supplemented with 10% fetal calf serum (FCS), 4 mM L-glutamine, 100 IU/mL of penicillin, and 100 µg/mL of streptomycin. 

The SARS-CoV-2 variants used in the study are the Zagreb-D614G strain [15] (GISAID: GR_hCoV-19/Croatia/ZG-297-20/2020) and the clinical isolate 0406*173 of the Delta (B.1.617.2) variant isolated at the Fran Mihaljevic clinical center in Zagreb (GISAID: hCoV-19/Croatia/6302/2021). Both SARS-CoV-2 isolates were propagated as previously described [16].

### 2.2. Monoclonal Antibodies

The STE90-C11 monoclonal antibody was previously described [10]. Shortly, STE90-C11 was identified in a phage display library from a convalescent patient as a binder to the RBD region and modified by silencing the Fc fragment. A functional Fc version of STE90-C11 with a modified murine receptor was generated for this study.

IgG FC05 [11] and IgG P17 [12] are monospecific antibodies binding to NTD and ACE2-RBD, respectively. The bispecific mAbs (CIY 1 + 1 and CIY 2 + 2) were generated by combining IgG FC05 and IgG P17 binding sites into bivalent (1 + 1) or tetravalent (2 + 2) bispecific eIg molecules as described previously [17], thus creating neutralizing antibodies against the RBD and the NTD regions of SARS-CoV-2 spike, as indicated in the Figure 7A.

### 2.3. Virus Stock Generation

SARS-CoV-2 variants were generated and virus stocks were quantified by plaque assay as described previously [16] with two key modifications. Firstly, CaCo2 cells (maintained in MEM supplemented with 5% FCS and 4 mM L-glutamine) were used for virus propagation and secondly, the virus stock was harvested 48 h post-infection (hpi). In brief, two T75 flasks were infected with an initial seeding stock. The virus stock supernatant was collected from all flasks at 48 hpi and spun at 3000× *g* for 10 min to remove cell debris. Then, the virus supernatant was concentrated using Vivaspin 20 concentrators (Sartorius Stedim Biotech, Goettingen, Germany) by spinning at 6000× *g* for 30 min. The resulting virus stock aliquots were stored at −80 °C until further use. Titration of viral stocks was performed as serial dilutions on Vero E6 cells cultured in virus titration media (VTM, DMEM supplemented with 5% FCS and 2 mM L-glutamine). The virus inoculum was added to VeroE6 cells and incubated at 37 °C. After 1 hour (h), the inoculum was removed, and the cells were overlaid with VTM supplemented with 1.5% carboxymethylcellulose (medium viscosity, C9481, Sigma-Aldrich, Burlington, MA, USA) and incubated at 37 °C for 3 days. 

Pseudotyped viruses were harvested as described before [18]. Briefly, HEK293T cells were transfected with expression plasmids (pCG1) encoding different S proteins of SARS-CoV-2 variants by using the calcium phosphate method. At 24 h post-transfection, the medium was removed, and cells were inoculated with a replication-deficient VSV vector lacking its glycoprotein and coding instead for an enhanced green fluorescent protein (GFP) (kindly provided by Gert Zimmer, Institute of Virology and Immunology, Mittelhäusern, Switzerland). Following 1 h incubation at 37 °C, the cells were washed with PBS, and culture media containing anti-VSV-G antibody (culture supernatant from I1-hybridoma cells; ATCC CRL-2700) were added. The pseudotype virus was harvested at 16–18 hpi; aliquots were stored at −80 °C.

### 2.4. Pseudovirus Neutralization Assay

Pseudovirus neutralization assays were performed as described previously in publications [19,20]. Monoclonal antibodies were serially diluted in VTM medium and mixed at a 1:1 ratio with pseudotyped particles. After 1 h incubation at 37 °C the mix was added to VeroE6 cells in 96-well plates. At 24 hpi, GFP expression was measured by using Incucyte S3 (Sartorius, Goettingen, Germany). Data were analyzed by Incucyte™ GUI Software (versions 2019B REV1 or 2021B).

### 2.5. Authentic Virus Neutralization Assay

The neutralization capacity of monoclonal antibodies on different cell lines (Vero E6, A549AT, and CaCo2) was tested using a 96-well plate format. Monoclonal antibodies were serial diluted and mixed with authentic SARS-CoV-2 variants. After 1 h incubation at 37 °C the mix was added to cells. Three days after incubation at 37 °C supernatant was collected and titrated on Vero E6 cells. Cells were freeze-thawed in a given volume of the corresponding culture medium and centrifuged for 5 min at 3000× *g* to remove cell debris. The collected supernatant was titrated on Vero E6 cells. Plaques were counted using the contrast phase of the Incucyte S3 (Sartorius, Goettingen, Germany).

### 2.6. Mouse Experiments and Organ Harvesting

K18-hACE2 (B6.Cg-Tg(K18-ACE2)2Prlmn/J) breeding pairs were purchased from Jackson Laboratory and maintained at the animal facility of the Helmholtz Center for Infection Research, Braunschweig. Virus and intranasal monoclonal antibody administration was performed under anesthesia with 80 mg Ketamin and 10 mg Xylazine. Nine- to twenty-six-week-old mice of both sexes were used for all experimental setups.

For the semi-therapeutic experiments, mice were treated intravenously with the indicated mAb STE90-C11 dose one hour before intranasal inoculation with 2 × 10^3^ PFU of the SARS-CoV-2 Delta variant. As for the antibody prophylaxis, mice were administrated with the indicated STE90-C11 dose by either intravenous or intranasal injection two days before SARS-CoV-2 infection.

Weight and health scores were monitored daily based on five criteria: spontaneous/social behavior, fur, fleeing behavior, posture, and weight loss, with a scale from no signs of symptoms (score = 0), mild and/or sporadic symptoms (score = 1), moderate and/or frequent symptoms (score = 2), to severe symptoms with a clear sign of heavy suffering (score = 3). Weight loss criterion was scored as follows: ≤1% (score = 0), 1–10% (score = 1), 10–20% (score = 2), and >20% (score = 3). Mice with a score of 3 in one criterion, or an overall score of ≥8, were removed from the experiments.

On day 5 post-SARS-CoV-2 infection, the indicated organs were harvested and homogenized in 500 or 1000 µL PBS with an MP Biomedical FastPrep 24 Tissue Homogenizer (MP Biomedicals, Irvine, CA, USA). Nasal wash was collected in 200 µL PBS. All samples were stored at −80 °C until further applications.

### 2.7. Viral Burden Measurement

RNA was isolated according to the manufacturer’s instructions (Rneasy RNA isolation kit, Qiagen). Eluted RNA was reverse transcribed and amplified using the TaqPath 1-step RT-qPCR Master Mix (Thermo Fischer Scientific, Carlsbad, MA, USA) and 2019-nCoV RUO kit (Integrated DNA Technologies (IDT), Coralville, IA, USA). For absolute viral RNA quantification, a standard curve was generated by serially diluting a SARS-CoV2 plasmid with the known copy numbers 200,000 copies/μL (2019-nCoV_N_Positive Control, #10006625, IDT, Coralville, IA, USA) at 1:10 ratio in all PCR analyses, with a quantitation limit of 2 copies of the plasmid standard in a single qPCR reaction. The viral RNA of each sample was quantified in duplicates and the mean viral RNA was calculated by the standard. RT-qPCR was performed using the StepOnePlusTM Real-Time PCR system (Thermo Fischer Scientific, Carlsbad, MA, USA) according to the manufacturer’s instructions.

### 2.8. Plaque Assay

Plaque assay was performed essentially as described [16] with the following modifications. Vero E6 cells were seeded at a density of 2 × 10^4^ cells per well in 96-well plate format. Organs were homogenized in 500 or 1000 µL PBS with an MP Biomedical FastPrep 24 Tissue Homogenizer (MP Biomedicals, Irvine, CA, USA) and centrifuged for 10 min at 3000× *g* at 4 °C. Homogenates were serially diluted with VTM medium, and cells were inoculated with 100 µL of the diluted homogenate and, incubated at 37 °C. After 1 h, diluted homogenate was replaced with 200 µL of VTM supplemented with 1.5% carboxymethylcellulose and incubated at 37 °C for 3 days. Plaque read-out was determined using the phase contrast of the Incucyte S3 microscope. 

### 2.9. Lung Histology and Immunofluorescence

Left lung tissue was formalin-fixed and paraffin-embedded (FFPE). The hematoxylin-eosin (HE) staining was performed according to standard laboratory procedures. A blinded and randomized evaluation was performed by a trained veterinary pathologist using a scoring system to assess the area affected by the pathological change, where a score of 1 = up to 30%, 2 = 40–70%, and 3 = more than 70%. The severity of the parameters, alveolar edema, interstitial pneumonia, and vasculitis, were graded as follows: 1 = mild, 2 = moderate, and 3 = severe. The scores for interstitial pneumonia and vasculitis were summarized to an inflammation score of 0–15. The presence of inflammatory cells was estimated for macrophages: 1 = occasionally seen, 2 = easily visible, large amounts, and 3 = dominating inflammatory cells. The scores for macrophages are qualitative markers and therefore, cannot be summarized in one score. 

Duplex immune-fluorescence staining was performed by staining for SARS-CoV-2 with mouse-anti nucleocapsid CoV-1/2 (Synaptic Systems, Goettingen, Germany, HS-452 11, clone 53E2, subtype: IgG2a) and for macrophages with rat-anti-mouse-MAC2 (Biozol Diagnostica, Eching, Germany/CEDARLANE, CL8942AP, clone M3/38). The slides were scanned with an Olympus VSI120 whole slide scanner using the Software VS-ASW 2.9.2 (Built 17,565). Scans were achieved with a 20× via (Maximum Intensity Projection) Z mode with 3 layers, and automatically analyzed with QuPath 0.32 (The University of Edinburgh, Edinburgh, Scotland, UK).

### 2.10. Statistics

All statistical analyses were performed as described in the indicated figure legends using GraphPad Prism 9. Statistical significance was determined using non-parametric one-way ANOVA, where Kruskal–Wallis was followed by Dunn´s multiple comparison post-test or one-way analysis of variance (ANOVA) followed by Dunnett´s multiple comparison post-analysis to compare the difference between three or more groups. Mann–Whitney test was used to compare results with two groups. The number of independent experiments used is indicated in figure legends.

## 3. Results

### 3.1. STE90-C11 Neutralizes SARS-CoV-2 Delta Variant In Vitro

We have shown previously that STE90-C11 binds to the RBD region of the SARS-CoV-2 spike and neutralizes the D614G strain of this virus [10]. STE90-C11 was predicted to neutralize the Delta variant (B.1.617.2) due to its ability to bind this variant, but this was not formally proven. Hence, we tested the in vitro neutralization capacity of STE90-C11 against the SARS-CoV-2 variants D614G (referred onward as WT) and Delta. Notably, STE90-C11 neutralized the Delta variant more efficiently than the WT (Figure 1). The 50% neutralization titer of the Delta variant was approximately 10-fold lower (Figure 1A), and a similar 10-fold difference was observed in the ability of antibodies to completely neutralize the virus (Figure 1B). We checked also the infectious virus titer intracellularly and it was neutralized completely in CaCo-2 and Vero E6 at a higher antibody concentration of 10 µg/mL (Appendix A), although the intracellular neutralization in A549AT cells was less efficient compared to the other two cell lines, CaCo-2 and Vero E6 (Appendix A).

### 3.2. STE90-C11 Protects against SARS-CoV-2 Delta Variant In Vivo

STE90-C11 controlled a D614G variant of SARS-CoV-2 highly efficiently when administered at a concentration of 30 mg/kg body weight [10]. Here we assessed the efficiency of STE90-C11 against the Delta variant in vivo. K18hACE2 mice were treated intravenously with a 30 mg/kg dose of STE90-C11 1 h before intranasal infection with 2 × 10^3^ PFU SARS-CoV-2 Delta (Figure 2A). Mice were monitored daily for weight loss and disease development. STE90-C11 treated mice maintained their weight (Figure 2B) compared to the control group and did not lose any body mass (Figure 2C). In addition, they were also protected from clinical signs throughout the infection (Figure 2D) and showed a low clinical score on day 5 post-infection (D5 pi) (Figure 2E). SARS-CoV-2 infection can cause respiratory and neurological diseases [21]. Therefore, we determined by plaque assay the viral burden in the lungs, trachea, and brain on D5 pi (Figure 2F). No infectious virus was detected in the lung and brain of treated mice compared to the untreated which showed high titer of virus in both organs. By contrast, in the trachea, no virus was detected even in the infected group. STE90-C11 neutralized the Delta variant to a value below our detection limit by D5 pi. To check STE90-C11 antiviral activity by a more sensitive assay, we measured viral RNA levels (Figure 2G). In the lungs, we observed a two-log reduction in viral RNA in treated vs. untreated mice, whereas in the trachea and brain, the viral copy number was completely reduced in the majority of mice. The presence of SARS-CoV-2 RNA can be detected in a broad range of tissues in both K18hACE2 mice and humans. Thus, we checked other organs (heart, spleen, and stomach) and nasal wash. We did not see any significant difference in the viral RNA upon STE90-C11 treatment in these organs, and we detected no viral RNA in the intestine (Appendix A). We also analyzed the lung pathology in hematoxylin-eosin (HE) stained sections that were scored for inflammatory alterations of the pulmonary structure, where lungs from STE90-C11 treated mice showed reduced inflammatory lesions (Appendix A). Immunohistochemistry revealed a decrease in SARS-CoV-2 nucleocapsid antigen and macrophages in the lungs of STE90-C11 treated mice compared to infected and untreated controls (Appendix A). The histological pneumonia or macrophage scores were on average reduced (Appendix A) but were not analyzed for statistical significance considering that the low number of tested samples would incur type II statistical errors. In summary, STE90-C11 protected efficiently against the Delta variant in vivo.

### 3.3. Prophylactic Efficacy of STE90-C11 against SARS-CoV-2 Delta Variant

To evaluate STE90-C11 in prophylactic settings, we treated K18hACE2 mice with a high dose (120 mg/kg) or a low one (30 mg/kg) of STE90-C11 intravenously two days before infection with SARS-CoV-2 Delta (Figure 3A). The health score and weight loss were monitored until D5 pi. Body mass was significantly reduced (by up to 20%) in untreated mice, whereas STE90-C11 treated mice showed no discernible weight loss (Figure 3B,C). Clinical scoring showed the same trend: treated mice showed lower clinical scores (Figure 3D) up to 5 days post-infection and a significant difference to the untreated and infected control group (Figure 3E). Virus titers, in the lung, trachea, and brain of treated mice were reduced to values below our detection limit (Figure 3F). SARS-CoV-2 viral RNA was 1000-fold reduced in the lung and 100,000-fold in the trachea of treated mice, although we did not see a significant reduction in nasal wash (Figure 3G). 

We next tested the prophylactic administration of STE90-C11 antibodies via intranasal injection. Therefore, we treated mice intranasally with either 30 mg/kg or 6 mg/kg of STE90-C11 two days prior to infection (Figure 4A). We used a lower STE90-C11 dose of 6 mg/kg as we assumed a higher neutralization efficiency upon topical antibody administration. Treated mice retained their weight throughout the infection (Figure 4B,C), while almost all of the infected control mice lost weight. Most of the treated mice showed low clinical scores, yet a few mice treated with 6 mg/kg of STE90-C11 showed moderate scores towards the end of the observation time (Figure 4D). Hence, clinical scores on the day of sacrifice were reduced in treated mice, but the difference was not significant in the group treated with the lower antibody dose (Figure 4E). Similarly, we observed a trend towards less complete protection with low STE90-C11 doses (6 mg/kg) in the lungs and brains (Figure 4F). A similar trend was also observed in the RNA viral burden. The intranasal and the intravenous prophylactic administration routes of STE90-C11 at the identical dose of 30 mg/kg showed no significant difference in body mass or clinical scores (Appendix A). Similarly, infectious viral loads and RNA viral burdens were not different between the two routes, except that the viral burden in the brain showed higher variability upon intranasal administration (Appendix A). Overall, the neutralization activity of STE90-C11 in vitro against the SARS-CoV-2 Delta variant was in line with the in vivo results. 

### 3.4. STE90-C11 Harboring a Functional Fc Fragment Elicits Better Protection against Delta Infection

The Fc fragment enables the interaction of antibodies with cell surface receptors on immune cells. This may lead to the activation of antibody-dependent phagocytosis or antibody-dependent killing of target cells. The role of FcR-mediated effector functions in therapeutic monoclonal antibodies has been discussed extensively [22,23,24]. STE90-C11 has a silenced Fc fragment to minimize the risk of side effects [10], such as antibody-dependent enhancement (ADE). However, silenced Fc fragment may also mean less antiviral efficiency. To check the efficiency of a non-silenced Fc, STE90-C11 was remodeled to a functional murine Fc fragment. Thereupon, we treated mice with either the STE90-C11 with no functional Fc fragment (NFc) or the functional one (FFC) before infection with SARS-CoV-2 Delta (Figure 5A). We chose for this experiment a low concentration of mAb (6, 2, and 0.6 mg/kg) to see if the functional Fc fragment may display an improved antiviral functionality. Most of the mice retained their weight upon infection, except for mice treated with 0.6 mg/kg of the NFc STE90-C11, where we observed a significant weight loss (Figure 5B,C). A similar trend was observed in the clinical score, whereby D5 pi we saw a shift of scores from low to moderate in mice receiving NFc STE90-C11, while FFc STE90-C11 provided more protection, especially at the highest concentration (Figure 5D,E). Virus titers in the lungs and brains of mice receiving higher doses (6 and 2 mg/kg) of FFc STE90-C11 were below the detection threshold, but this was not the case for most of the mice receiving the same dose of NFc STE90-C11 (Figure 5F). Low-dose NFc STE90-C11 (0.6 mg/kg) was less efficient in controlling the virus, but an FFc was again performing better than NFc STE90-C11. Viral RNA levels were only marginally reduced in the lungs and trachea in comparison to the infected control group, except for the highest concentration of FFC STE90-C11, where the effects were more easily discernible (Figure 5G). However, we observed a substantial reduction in virus RNA loads in the brains of mice treated with FFc STE90-C11 at all concentrations and with the highest concentration of 6 mg/kg of NFc STE90-C11. 

Our data argued that STE90-C11 neutralizes the SARS-CoV-2 Delta variant and protects mice against severe disease development at higher and lower doses. Different routes of administration had similar efficiency in protection, but a functional Fc receptor improved the antiviral activity, especially at lower antibody concentrations.

### 3.5. Neutralization of Escape Mutants by STE90-C11

Since November 2021 the SARS-CoV-2 variant “Omicron” has spread through the world and outcompeted previously dominant variants, including the Delta variant. While Delta infections elicited a higher mortality rate, Omicron sublineages have repressed them due to better replicative properties in a predominantly immune population, thus expanding in a multitude of subvariants with novel mutations [25]. We tested therefore the ability of STE90-C11 to neutralize the Omicron variant by pseudovirus neutralization assays. We used pseudoviruses harboring either the SARS-CoV-2 spike glycoprotein of the original Wildtype strain (B.1), Beta (B1. 351), Delta (B1.617.2), or Omicron (BA.1) on different cell lines. We observed a complete loss of neutralization of STE90-C11 against Beta and Omicron variants in all cell lines (Figure 6A). Similarly, we observed a lack of control of the Alpha variant (B.1.1.7) by STE90-C11. Therefore, at least some of the mutations present in these variants provided the virus with the ability to avoid recognition by STE90-C11 antibodies, while retaining the ability to infect target cells. Thus, to characterize this phenomenon in more depth, we generated a library of mutant pseudoviruses that express recombinant spike proteins that carry any single point mutation that is present in the Omicron BA.1–BA-5 variants, but not in the wild-type parental variants [26]. The point mutations were divided according to their location on the spike protein; N-terminal domain (NTD), receptor binding domain (RBD), the receptor binding motif (RBM) within the RBD, spike subdomains 1 and 2 (SD1/2), and finally on the spike 2 subunit (Figure 6B). Mutations within the NTD, SD1/2, and spike 2 were completely neutralized at higher STE90-C11 concentrations (Figure 6C). However, we observed a failure of neutralization against several mutations located in the RBD/RBM region of the spike protein. The mutations K417N, and N501Y showed a complete escape of recognition from STE90-C11, while S375F, Y505H, and the mutations at position 371 showed a clear, but incomplete loss of neutralization (Figure 6D). These results were consistent with our previous data that STE90-C11 poorly binds to the Beta variant due to the K417N mutation [10], but also argued that the other independent mutations facilitated the escape of recognition by this monoclonal antibody and thus the loss of its neutralizing capacity. 

### 3.6. Neutralization of SARS-CoV-2 Variants by Bispecific Antibodies

While these results argued that a single amino acid change may result in losses in neutralization capacity and immune escape from monoclonal antibodies, they also imply that combinations of monoclonal antibodies may be more resilient to virus mutations. Bispecific antibodies (BsAbs) binding to two different antigens or two different epitopes of the same antigen may therefore be more efficient and resilient to viral immune escape [27,28]. We used our recently described BsAbs format [17] to generate bispecific antibodies targeting the NTD and the RBD and used these in neutralization assays against Omicron or our library of pseudoviruses bearing the SARS-CoV-2 S point mutations. Two parent IgGs P17 and FC05 have been structured in a way that their Fab fragments bind to the ACE2-RBD region and the NTD regions, respectively (Figure 7A) and combined into BsAbs CIY 1 + 1 and CIY 2 + 2, as combinations of these parental IgGs that may simultaneously bind to RBD and the NTD. Neutralizing capacity was tested in a pseudovirus assay with spike glycoproteins of the WT, Alpha, Beta, Delta, and Omicron variants. We observed that IgG P17 efficiently neutralized the SARS-CoV-2 variants WT, Alpha, and Delta, but not the Beta or Omicron (Figure 7B). On the other hand, IgG FC05 neutralized in part the SARS-CoV-2 WT strain at higher concentrations. While CIY 1 + 1, which carried one Fab fragment with the P17 and one with the FC05 specificity behaved akin to the FC05 and neutralized only the WT variant, the CIY 2 + 2, endowed with 2 Fabs of each specificity behaved similarly to the P17 monoclonal and acted against the same three variants. We checked the antibodies’ efficiency also with the authentic SARS-CoV-2 variant WT and Delta with similar results (Figure 7C). We surmised that the epitope recognized by P17 must be shared by the Beta and the Omicron variant, but to validate this assumption, we tested the ability of the P17 Ab to neutralize the pseudoviruses from our point mutation library. The monovalent Ab IgG P17 neutralized efficiently all point mutations except the E484A mutation in the RBM region (Figure 7D), which is known to be present in the Omicron and the Beta variant. Intriguingly, variants with the 371 mutations were also imperfectly controlled, such as in the case of STE90-C11, which may be a general effect of point mutations at this location on virus growth, as the same effect was also observed in the bivalent mutant or even in presence of polyclonal sera [26]. We wondered if the CIY 2 + 2 BsAb, which also failed to neutralize the Omicron library, and the Beta variant would fail to neutralize the E484A mutant from the library, despite the presence of the FC05 Fab. To our surprise, all mutants, including the E484A were controlled, though the efficiency was somewhat decreased for the mutation E484A (Figure 7D). These results argue that multivalent bispecific antibodies, such as CIY 2 + 2, have the potential to neutralize viruses with any point mutation present in a highly divergent variant, such as the Omicron, but complex mutations in a rapidly evolving variant, such as Omicron, or the recently emerged BA.2.86 (pirola) may still escape recognition of complex monoclonal preparations.

## 4. Discussion

Most monoclonal antibodies developed and approved as anti-SARS-CoV-2 therapeutics have lost their efficiency against the Omicron variant and its subsequent derivatives [29,30,31]. Most of the mAbs against SARS-CoV-2 have been developed to target the RBD region of the spike protein and the virus’ ability to bind to cells, but this region of the virus proved to mutate exceptionally rapidly, likely to confer to the virus a selective advantage in a seropositive population. Similarly, the mAb STE90-C11, which was previously shown to protect against SARS-CoV-2-D614G [10], is shown here to neutralize the Delta variant more efficiently than the ancestral virus in vitro and in vivo but fails to neutralize the Omicron variant. Indeed, testing this mAb intravenously in vivo in K18hACE2 mice, we observed complete protection of the mice with a concentration of 30 mg/kg. Viral titers and load decreased up to ~10^10^-fold in the lung, trachea, and brain. Surprisingly, we did not see any difference in other organs in the treated mice. It seems that the monoclonal antibody did not reach the peripheral organs. According to previous studies [32,33,34], in K18hACE2 mice, Delta infection leads to high inflammation in the lungs. This was reduced here in the lungs when treating mice with STE90-C11. 

The intranasal administration of STE90-C11 acted as efficiently as the intravenous route in preventing COVID-19 symptoms and reducing the viral load in organs. Previous studies conducted by Hawle et al. and Lu et al. have shown that administering monoclonal antibodies through nasal delivery can protect against SARS-CoV-2 infection [35,36]. Lu et al. found that intranasal administration of two neutralizing antibodies (F61 and H121) was highly effective in protecting against the Delta and Omicron variants in a prophylactic setting [36]. Similarly, the monoclonal antibody DZIF-10c by Hawle et al. showed efficient neutralization of SARS-CoV-2 variants and reduced lung pathology when applied topically and prophylactically [35]. Additionally, an earlier study conducted by Piepenbrick in 2021 suggested that nebulized monoclonal antibody 1212C2, when inhaled, could significantly reduce viral burden and lung pathology in a hamster model [37]. Many studies have tested antibodies with modified Fc fragments in monoclonal antibodies to assess their functionality and the lifespan of these molecules [23,38,39]. The original STE90-C11 formulation was based on a silenced Fc fragment to avoid potential antibody-dependent cell-mediated reactions. However, STE90-C11 with the Wildtype (non-silenced) Fc fragment showed no increase in pathogenicity and protected the mice at much lower doses than STE90-C11 with the silenced Fc, controlling the virus more efficiently in the brain. This might be explained by the activation and recruitment of antiviral immune cells by the antibodies with the functional Fc fragment, but the exact effects and cell types involved have not been addressed within the scope of the current study. 

One limitation of our in vivo work is the duration of the experimental design; longer post-infection monitoring of the mice would have allowed us to define if a single dose of antibodies is sufficient to prevent late-onset symptoms. Another limitation was that we did not assess the efficiency of STE90-C11 administered after virus infection, but only in prophylactic or semi-therapeutic approaches. Nevertheless, our data show that STE90-C11 provides in vivo protection against the Delta variant regardless of the administration route and that functional Fc fragments improved the antiviral activity of the STE90-C11 antibodies. Here again, it is imperative to verify the safety implications of the functional Fc Fragment. 

In this study, we mapped the mutations that are present in the Omicron variant and observed that K417N and N501Y, substantially contributed to immune escape, while the E484K mutation was not relevant for the antiviral effect of STE90-C11. Hence it is not surprising that the Alpha variant (B1.1.7), with the N501Y mutation as the only one in the RBD, was poorly controlled. The other two mutations of B1.1.7, 69/70 deletion, and P681H near the S1/S2 cleavage site were properly neutralized. The loss of neutralization of Omicron was therefore due to the poor recognition of K417N, and N501Y, and to the partial neutralization of G496S, and Y505H. Based on these data, we confirmed some of the mutations (K417N and N501Y) described previously [10] but we also suggest other mutations such as (G496S, and Y505H) may lead to loss of neutralization of STE90-C11. Therefore, since any of these mutations in isolation impaired spike recognition and virus control by STE90-C11, the data argued that this antibody recognized a complex structural epitope whose morphology depended on all of these residues. 

Bispecific antibodies (BsAb) were originally generated against carcinoma and diabetes [28]. In the last years, the studies describing BsAbs against SARS-CoV-2 have increased [27,40,41]. To validate if the library of Omicron point mutations may be used for mapping monoclonal antibody responses more generally, we tested the mAb P17 recognizing the SARS-CoV-2 S RBD and the BsAb CYI 2 + 2, which contained the P17 Fab fragment and a Fab recognizing the NTD region of the S protein [17]. While the monovalent IgG P17 showed a loss of neutralization against the E484A mutant, the BsAb CIY 2 + 2 showed no absolute loss of neutralization for any mutations. While we did not identify the additional mutations in the Omicron variant that were responsible for the loss of neutralization by the BSAb CIY 2 + 2, our data show that the point mutation that was critical for the neutralization by the P17 mAbs was not sufficient for the escape of the bispecific antibody that combined the P17 and the FC05 Fab fragments. 

Hence, we determined the single mutations that were responsible for the neutralization loss of STE90-C11 and P17 against SARS-CoV-2 variants but also showed that BsAbs can neutralize all individual variants in a library due to complementation effects.

## Figures and Tables

**Figure 1 viruses-15-02153-f001:**
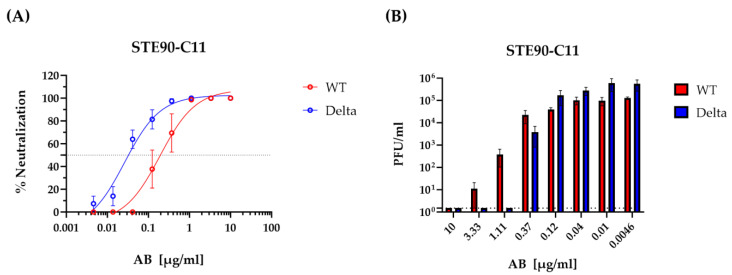
Neutralization of SARS-CoV-2 variant strains by the mAb STE90-C11. Authentic SARS-CoV-2 neutralization titration by STE90-C11 performed using Vero E6 cells against SARS-CoV-2 WT-D614G (red) and Delta (blue) are represented as percentage of neutralization (**A**) and plaque number per ml (**B**). Lines represent nonlinear regression fit and data were shown as mean ± SEM of two independent experiments with two to three technical replicates.

**Figure 2 viruses-15-02153-f002:**
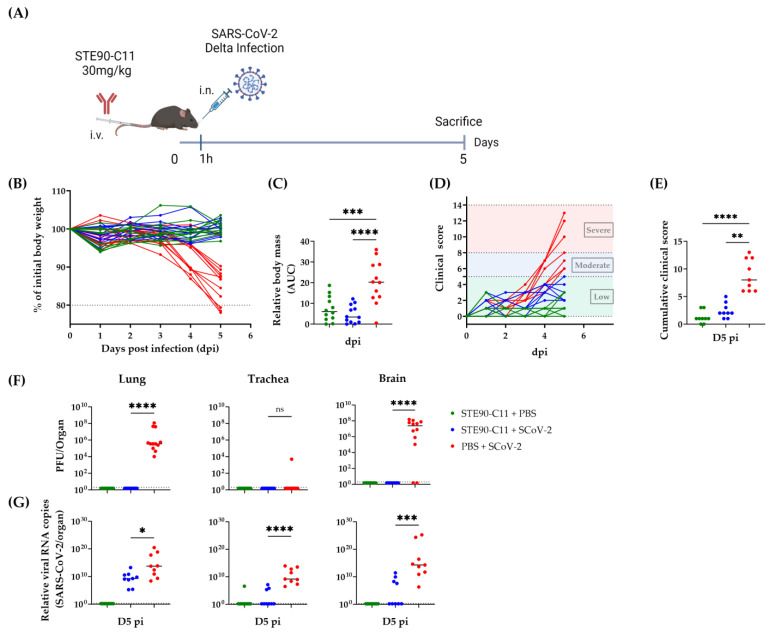
STE90-C11 semi-therapeutic application against SARS-CoV-2 Delta (**A**) Experimental setup: 9–18 week-old mice received 30 mg/kg of STE90-C11 by intravenous injection (i.v.) 1 h before intranasal infection (i.n.) with 2 × 10^3^ PFU SARS-CoV-2 Delta strain in 20 µL volume. Organs were collected 5 days later. (**B**) Weight change after infection with SARS-CoV-2. (**C**) Cumulative relative mass reduction in SARS-CoV-2 infected and with mAb treated mice until D5 pi are shown as area under the curve (AUC). (**D**) Daily clinical scores upon SARS-CoV-2 infection and mAb treatment. The indicated thresholds represent the clinical severity of mice; low (green), moderate (blue), and severe (red, humane end-point). (**E**) Cumulative clinical score on D5 pi. (**F**) SARS-CoV-2 viral load at D5 pi in the lung (left), trachea (middle), and brain (left). (**G**) Viral RNA levels in the lung (left), trachea (middle), and brain (left) were measured. The dotted line indicates the limit of detection of the assay. Pooled data (n = 9–12 per group) from four independent experiments are shown. Each symbol is an individual mouse, and horizontal lines indicate the median of biological replicates. Statistical significance versus the infected control group was calculated using (**C**) one-way analysis of variance (ANOVA) followed by Dunnett´s post-analysis, (**E**,**F**) Kruskal–Wallis test followed by Dunn´s post-analysis, and (**G**) Mann–Whitney test. * for *p* < 0.05, ** for *p* < 0.005, *** for *p* < 0.001 and **** for *p* < 0.0001. ns: non-significant.

**Figure 3 viruses-15-02153-f003:**
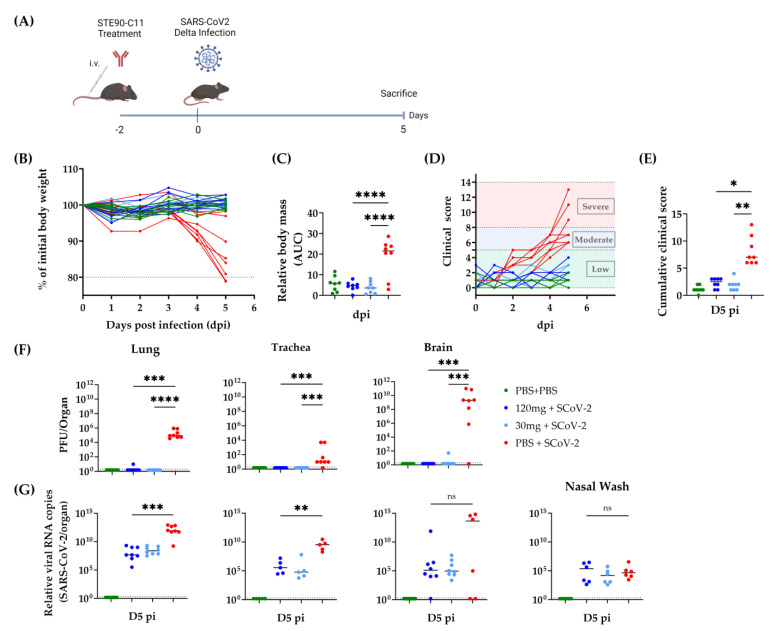
STE90-C11 prophylaxis against SARS-CoV-2 Delta. (**A**) Experimental setup: 10–18 week-old female and male K18-hACE 2 transgenic mice received 120 mg/kg, 30 mg/kg of STE90-C11 or PBS by intravenous injection two days upon intranasal inoculation with 2 × 10^3^ PFU SARS-CoV-2 Delta strain in 20 µL volume. Organs were collected 5 days later. (**B**) Relative weight upon infection in single mice. (**C**) Cumulative relative mass reduction in individual mice represented as area under the curve (AUC). (**D**) Daily clinical scores upon infection and mAb treatment. Clinical severity thresholds are indicated as low (green), moderate (blue), and severe (red, humane end-point). (**E**) Cumulative clinical score on D5 pi. (**F**) SARS-CoV-2 viral loads at dpi 5 in lungs, trachea, and brain. (**G**) Viral RNA levels in the lung, trachea, brain, and nasal wash were measured. The dotted line indicates the limit of detection of the assay. Data from three independent experiments were pooled (n = 6–8 per group). Each symbol is an individual mouse, and horizontal lines indicate the median of biological replicates. Statistical significance versus the infected control group was calculated using (**C**) One-way analysis of variance (ANOVA) followed by Dunnett´s post-analysis or (**E**,**F**) Kruskal–Wallis test followed by Dunn´s post-analysis, and (**G**) Mann–Whitney test. * for *p* < 0.05, ** for *p* < 0.005, *** for *p* < 0.001 and **** for *p* < 0.0001. ns non-significant.

**Figure 4 viruses-15-02153-f004:**
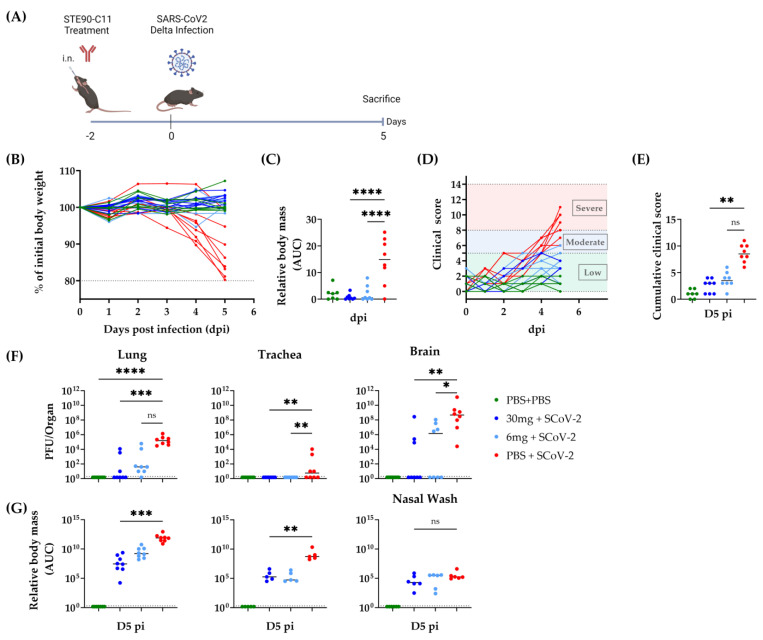
Intranasal application of STE90-C11 against SARS-CoV-2 Delta (**A**) Experimental setup: 10–18 week-old female and male K18-hACE 2 transgenic mice received 30 mg/kg, 6 mg/kg of STE90-C11 or PBS by intranasal injection two days upon infection with 2 × 10^3^ PFU SARS-CoV-2 Delta strain in 20 µL volume intranasal. Organs were collected at 5 days post-infection. (**B**) Weight change after infection with SARS-CoV-2. (**C**) Cumulative relative mass reduction in SARS-CoV-2 infected and with mAb treated mice until D5 pi are shown as area under the curve (AUC). (**D**) Daily clinical scores upon SARS-CoV-2 infection and mAb treatment. The indicated thresholds represent the clinical severity of mice; low (green), moderate (blue), and severe (red, humane end-point). (**E**) Cumulative clinical score on D5 pi. (**F**) SARS-CoV-2 viral load at D5 pi in the lung (left), trachea (middle), and brain (left). (**G**) Viral RNA levels in the lung, trachea, and nasal wash were measured. The dotted line indicates the limit of detection of the assay. Pooled data (n = 3–6 per group) from three independent experiments are shown. Each symbol is an individual mouse, and horizontal lines indicate the median of biological replicates. Statistical significance versus the infected control group was calculated using (**C**) one-way analysis of variance (ANOVA) followed by Dunnett´s post-analysis (**E**,**F**) Kruskal–Wallis test followed by Dunn’s post-analysis, and (**G**) Mann–Whitney test. * for *p* < 0.05, ** for *p* < 0.005, *** for *p* < 0.001 and **** for *p* < 0.0001. ns non-significant.

**Figure 5 viruses-15-02153-f005:**
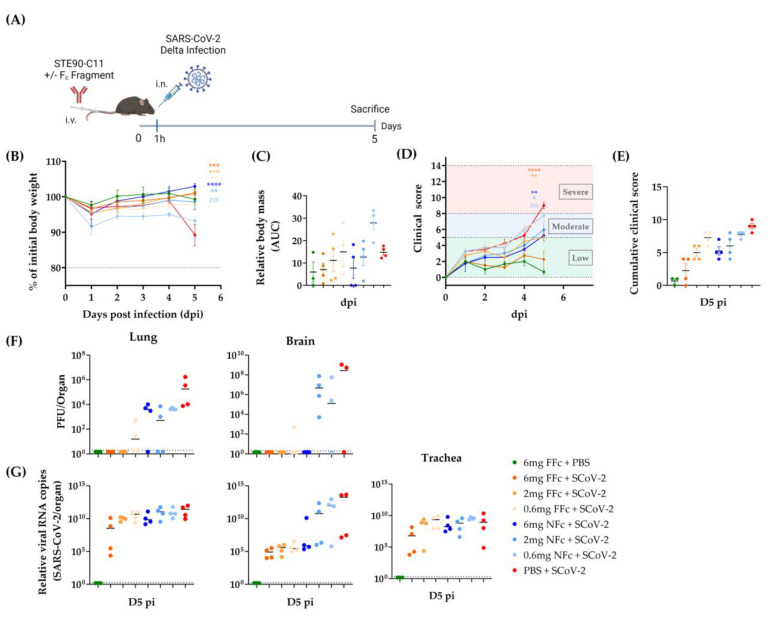
Contribution of functional Fc fragment for the function of STE90-C11 against SARS-CoV-2 Delta infection. (**A**) Experimental setup: 13–25 week-old female and male K18-hACE 2 transgenic mice received different concentrations of either the functional Fc STE90-C11 (orange) or the non-functional Fc STE90-C11 (blue) by intravenous injection one hour before intranasal inoculation with 2 × 10^3^ PFU SARS-CoV-2 Delta strain in 20 µL volume. Organs were collected at 5 days post-infection. (**B**) Weight change after infection with SARS-CoV-2. (**C**) Cumulative relative mass reduction in SARS-CoV-2 infected and with mAb treated mice until D5 pi are shown as area under the curve (AUC). (**D**) Daily clinical scores upon SARS-CoV-2 infection and mAb treatment. The indicated thresholds represent the clinical severity of mice; low (green), moderate (blue), and severe (red, humane end-point). (**E**) Cumulative clinical score on D5 pi. (**F**) SARS-CoV-2 viral load at D5 pi in lung (left) and brain (left). (**G**) Viral RNA levels in the lung (left), brain (middle), and trachea (left) were measured. The dotted line indicates the limit of detection of the assay. Pooled data (n = 4 per group) from two independent experiments are shown. Each symbol is an individual mouse, and horizontal lines indicate the median of biological replicates. * for *p* < 0.05, ** for *p* < 0.005, *** for *p* < 0.001 and **** for *p* < 0.0001. ns non-significant.

**Figure 6 viruses-15-02153-f006:**
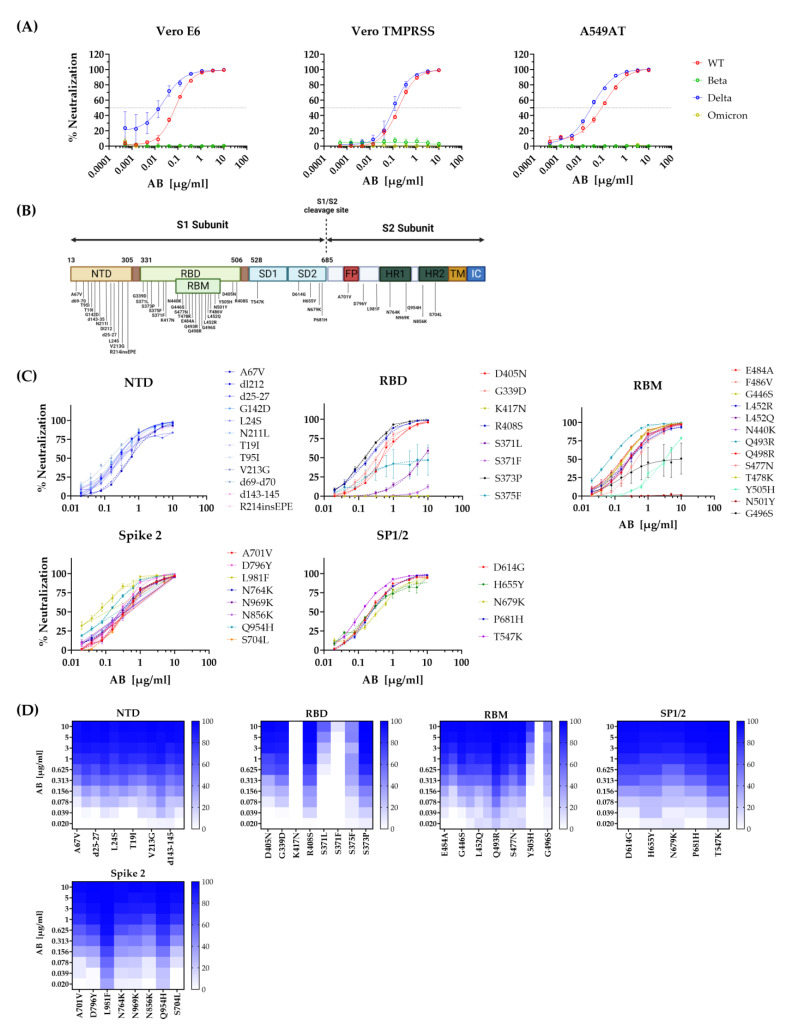
Single mutation mapping of STE90-C11 in vitro. (**A**) The neutralization capacity of STE90-C11 was assessed using pseudoviruses expressing SARS-CoV-2 spike glycoprotein of either the original WT strain (red), Beta (green), Delta (blue), or Omicron (yellow) at different antibody concentrations on different cell types. (**B**) Schematic overview of the Omicron single mutations tested. (**C**) STE90-C11 neutralization capacity against pseudoviruses harboring single mutations of the Omicron variant. Data were divided after the region single mutations were located: NTD, RBD, RBM, SP1/2, and Spike 2. Lines represent nonlinear regression fit and data were shown as mean ± SEM of two independent experiments with two to three technical replicates. (**D**) The Heatmap-like representation shows the neutralization of STE90-C11 against Omicron single mutations in a dose-dependent manner.

**Figure 7 viruses-15-02153-f007:**
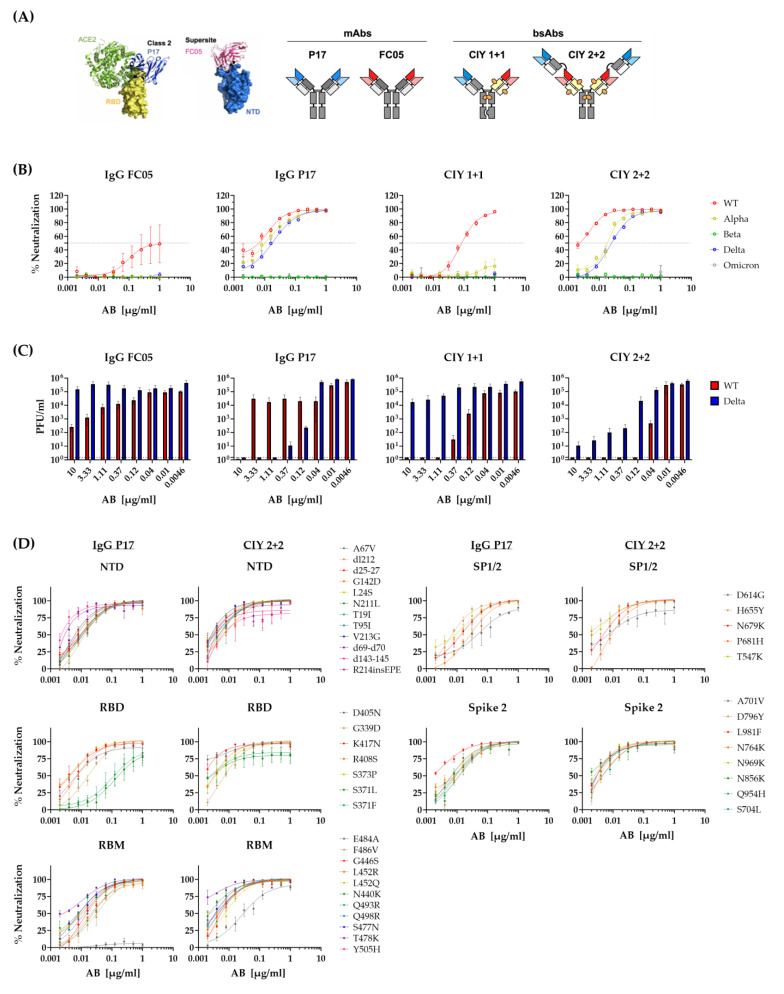
Neutralization of SARS-CoV-2 variants of concern by the bispecific monoclonal antibodies. (**A**) Bispecific mAbs structures and regions of binding on the SARS-CoV-2 spike protein. (**B**) Neutralization capacities were assessed using pseudoviruses expressing SARS-CoV-2 spike glycoprotein of either the original WT strain (red), Alpha (yellow), Beta (green), Delta (blue), or Omicron (gray) at different antibody concentrations. (**C**) Authentic SARS-CoV-2 neutralization titration by the bispecific mAbs performed using Vero E6 cells are represented as plaque number per ml. Lines represent nonlinear regression fit and data were shown as mean ± SEM of two independent experiments with two to three technical replicates. (**D**) Neutralization capacity of IgG P17 (left) and CIY 2 + 2 (right) against pseudoviruses harboring single mutations of the Omicron variant. Data were divided after the region single mutations were located: NTD, RBD, RBM, SP1/2, and Spike 2. Lines represent nonlinear regression fit and data were shown as mean ± SEM of two independent experiments with two to three technical replicates.

## Data Availability

All relevant data is available in the manuscript.

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
