# Peer review of "Evaluation of the Neutralizing Antibody STE90-C11 against SARS-CoV-2 Delta Infection and Its Recognition of Other Variants of Concerns"

_viruses, 2023, doi:10.3390/v15112153_

Round 1

Reviewer 1 Report

dear authors, in light of SARS-CoV-2 with its variants still representing a health problem worldwide, your work is very interesting and comprehensively analyzed. the optimized STE90-C11 mAb and/or its humanized bispecific will represent a promising avenue to control some variants of SARS-CoV-2

Thank you,

Author Response

We thank the reviewer for their positive review. 

Reviewer 2 Report

In this manuscript, Abassi and colleagues evaluated the neutralizing effects in vitro and prophylactic abilities in vivo of a previously generated mAb STE90-C11. They found that the mAb STE90-C11 could neutralize the SARS-CoV-2 Delta variant but not Beta and Omicron variants. They further identified the mutation sites responsible for escaping recognition by STE90-C11. This work was well designed and conducted. However, after reviewing, I have several concerns about this manuscript.

Major:

1. As we all know, SARS-CoV-2 is evolving so rapidly that the circulating strain is no longer Delta but Omicron. What is needed are antibodies against the strains that are circulating now or broad spectrum neutralizing antibodies. The mAb STE90-C11 in this study can only neutralize the WT and Delta variants. So it is obvious that such research has limited value and significance for the current and future prophylactic and therapeutic research on COVID-19.

2. In “Section 3.6. Neutralization of SARS-CoV-2 variants by bispecific antibodies.” The content and results of this part are not strongly related to the whole paper, and the two antibodies selected to construct bispecific antibodies do not have neutralizing activity against Omicron variants. I suggest this part be deleted.

Minor:

1. Fig 2 (B to G), Fig 3 (B to G), Fig 5 (B to G), Fig 6. The font size of these figures is too small to see clearly. It is suggested to adjust the font size of the pictures.

Author Response

We thank the reviewer for the overall positive comments. To the specific issues raised: 

Major:

1. We agree with the reviewer that the STE90-C11 antiviral effects are only applicable to COVID variants that have been repressed by emerging variants and hence of little practical use, which was a main focus of our paper. This was also the impetus for the work on the mapping of immune evasion properties of Omicron against this antibody, to understand which mutations may revert the phenotype and render the monoclonal antibody functional again. 
2. Our mapping library identified aminoacid residues that were recognized by the STE90-C11 antibody and whose mutation was sufficient to impair neutralization (417K and 501Y). Hence, mutations corresponding to single aa exchanges were sufficient to block the antiviral activity of a monoclonal antibody, but it remained unclear if bivalent antibodies would be affected by the mutation of only one of the redundant epitopes. It was possible that the two divergent Fab fragments would have additive effects, which would result in a partial loss of neutralization, or that the would act redundantly which would entirely kill abrogate virus control. Hence, we used a bivalent antibody that was know to us as failing to control omicron, but controling the wild type virus. We showed that all of the viruses with the single mutations could be controlled, thus showing that the effects of such bivalent constructs are redundant and not additive. We think that this information is relevant for the readers and fitting the overall scope of the manuscript. 

Minor:

1. We agree with the reviewer and have enlarged the fonts of the figures that have been mentioned.